# Substituted Syndecan-2-Derived Mimetic Peptides Show Improved Antitumor Activity over the Parent Syndecan-2-Derived Peptide

**DOI:** 10.3390/ijms23115888

**Published:** 2022-05-24

**Authors:** Bohee Jang, Ayoung Kim, Yejin Lee, Jisun Hwang, Jee-Young Sung, Eun-Ju Jang, Yong-Nyun Kim, Ji-Hye Yun, Jeongmin Han, Ji-Joon Song, Weontae Lee, Eok-Soo Oh

**Affiliations:** 1Department of Life Sciences, College of Natural Sciences, Ewha Womans University, Seoul 03760, Korea; bhjang@ewha.ac.kr (B.J.); aykim1209@ewhain.net (A.K.); oj3388@naver.com (Y.L.); jisunhwang@ewha.ac.kr (J.H.); 2Division of Translational Science, National Cancer Center, 323 Ilsan-ro, Ilsandong-gu, Goyang-si 10408, Korea; sungjy@ncc.re.kr (J.-Y.S.); eunju2190@ncc.re.kr (E.-J.J.); ynk@ncc.re.kr (Y.-N.K.); 3PCG-Biotech, Ltd., Yonsei Engineering Research Park 114A, Yonsei University, Yonsei-ro 50, Seoul 03722, Korea; jihye2@spin.yonsei.ac.kr (J.-H.Y.); jmhan723@yonsei.ac.kr (J.H.); songj@kaist.ac.kr (J.-J.S.); 4Department of Biochemistry, College of Life Science and Biotechnology, Yonsei University, Seoul 03722, Korea

**Keywords:** colon cancer, matrix metalloproteinase-7, syndecan-2, therapeutic agent

## Abstract

We previously showed that a synthetic peptide (S2-P) corresponding to a portion of the human syndecan-2 (SDC2) sequence can bind to the pro-domain of matrix metalloproteinase-7 (MMP-7) to inhibit colon cancer activities. Since S2-P had a relatively weak binding affinity for the MMP-7 pro-domain, we herein modified the amino acid sequence of S2-P to improve the anticancer potential. On the basis of the interaction structure of S2-P and MMP-7, four peptides were generated by replacing amino acids near Tyr 51, which is critical for the interaction. The SDC2-mimetic peptides harboring an Ala-to-Asp substitution at the C-terminal side of Tyr 51 (S2-D) or with an Ala-to-Phe substitution at the N-terminal side of Tyr 51 and an Ala-to-Asp substitution at the C-terminal side of Tyr 51 (S2-FE) showed improved interaction affinities for the MMP-7 pro-domain. Compared to S2-P, S2-FE was better able to inhibit the SDC2–MMP-7 interaction, the cell surface localization of MMP-7, the gelatin degradation activity of MMP-7, and the cancer activities (cell migration, invasion, and colony-forming activity) of human HCT116 colon cancer cells in vitro. In vivo, S2-FE inhibited the primary tumor growth and lung metastasis of CT26 mouse colon cancer cells in a xenograft mouse model. Together, these data suggest that S2-FE could be useful therapeutic anticancer peptides for colon cancer.

## 1. Introduction

The syndecans (SDCs) comprise a family of highly conserved heparan sulfate proteoglycans that act as cell surface receptors; they bind to the extracellular matrix (ECM) and thereby regulate cell adhesion and related cell functions [1,2]. Although there is some functional redundancy, all syndecan family members play distinct functional roles. Among them, SDC2 has specific functions in colorectal cancer [3]. Previous studies have shown that SDC2 is highly expressed in colon cancer cell lines [4,5] compared to normal cell lines and in cancer tissues [6] compared to neighboring normal tissues, and that this plays an important role in regulating colon cancer activities [3,7]. As seen for other heparan sulfate proteoglycans, SDC2 communicates with the ECM via an extracellular heparan-binding domain [8]. This domain of SDC2 directly interacts with ECM proteins, such as fibronectin [9] and integrin α2 [10] in order to regulate intracellular signaling. Through this regulation, increased SDC2 promotes the cell adhesion and spreading of colon cancer cells [4]. SDC2 also provides docking sites for various ligands, such as growth factors and proteases, to regulate extracellular events. For example, matrix metalloproteinase-7 (MMP-7) binds to the extracellular domain of SDC2 and subsequently undergoes activation in colon cancer cells [11].

The MMPs are a large family of calcium-dependent zinc-containing endopeptidases that are responsible for tissue remodeling and ECM degradation [12]. This proteolytic activity of MMPs is crucial for the ability of cancer cells to exhibit increased cell growth, metastasis, and angiogenesis [13]. Therefore, many efforts have been made to develop agents that can inhibit the proteolytic activity of MMPs for use as anticancer drugs. Early efforts to develop MMP inhibitors mainly focused on compounds that mimicked the natural peptide substrate of the desired MMP by containing a catalytic zinc ion and a group that can chelate the backbone [14,15]. Although hydroxamate-based chelation inhibitors (e.g., batimastat) showed promising antitumor effects in animal models of various cancers, clinical trials have been unsuccessful due to severe side effects.

MMP-7 is the smallest MMP; its expression is associated with the development and metastasis of colon cancer [16,17]. MMP-7 and the other MMPs are synthesized and secreted as a pro-enzyme (zymogen), and their cleavage-mediated activation is essential for their proteolytic activity. MMP-7 (and the other MMPs) contains two domains, the pro-domain and the catalytic domain. The pro-domain has free cysteine residues that interfere with the activity of latent pro-MMP-7 by interacting with a zinc ion in the catalytic domain, which exhibits enzymatic activity upon cleavage of the pro-domain [18,19]. The active form of MMP-7 degrades an ECM complex that includes a heparan sulfate proteoglycan [20] and perlecan [21] to promote tumor cell migratory phenotypes and metastasis. MMP-7 also cleaves cell surface receptors, such as E-cadherin, to promote cancer cell invasion [22,23,24].

In addition to their pro-domain-cleavage-mediated activation, the cell surface localization of MMPs is an important step for their activation and substrate access. For instance, pro-MMP-2 localizes to the cell surface by interacting with the membrane MMP, MT1-MMP, and tissue inhibitor of metalloproteinase-2 (TIMP-2). On the cell surface, pro-MMP-2-TIPM-2-MT1-MMP induces cleavage of the MMP-2 pro-domain, leading to the activation of MMP-2 [25]. Another surface-localized MMP is MMP-1, which localizes at the front of the invasive edge of the cancer cell and cleaves ECM components, such as type I collagen, in human lung carcinoma [26] and pancreatic cancer [27].

A previous study showed that SDC2 acts as a cell surface docking receptor of MMP-7 [6]. In the ECM, secreted pro-MMP-7 binds to the extracellular domain of SDC2 core protein at the cell surface [11]. Tyrosine residue 51 (Tyr 51) of the extracellular domain of SDC2 occupies the binding pocket of pro-MMP-7, which is formed by the α2 helix-loop-α3 loop of the MMP-7 pro-domain; through this direct interaction, pro-MMP-7 is converted to active MMP-7 [11]. Thus, the SDC2–MMP-7 interaction appears to be important for the enzymatic activity of MMP-7 in colon cancer cells. Interestingly, a synthetic peptide (S2-P) corresponding to a portion of the SDC2 sequence bound the pro-domain of MMP-7 to interrupt the SDC2–MMP-7 interaction and block the activation of pro-MMP-7, and this inhibited the tumorigenic activities of colon cancer cells [11]. These results suggested that MMP-7 activity could be decreased by inhibiting its cell surface localization (and thereby its activation), and that S2-P could potentially act as an anticancer peptide. In this context, we sought to further investigate the potential of S2-P-derived SDC2-mimetic peptides as anticancer therapeutics that could potentially replace hydroxamate-based chelating inhibitors.

## 2. Results

### 2.1. Syndecan-2 Mediated the Localization of Pro-MMP-7 by Interacting with Pro-Domain of MMP-7

We have previously reported that that the N-terminal extracellular domain of SDC2 interacts with MMP-7 through its pro-domain [6]. In this study, we further determined whether this interaction is sufficient for cell surface localization of MMP-7 (Figure 1). As previously reported [28], human HT29 colon adenocarcinoma cells stably expressing SDC2 (HT29-SDC2) showed upregulation of the mRNA expression of MMP-7 (Appendix A), and their conditioned media showed increased MMP-7 enzymatic activity to cleave an exogenous substrate (Mca-Pro-Leu-Gly-Leu-Dap(Dnp)-Ala-Arg-NH2, Mca-PLA-Nva-Dap(Dnp)-AR-NH2) (Figure 1A). Moreover, HT29-SDC2 cell showed increased cell surface localization of MMP-7 (Figure 1B), confirming that SDC2 regulates the cell surface localization of MMP-7 in colon cancer cells.

To further investigate whether SDC2 directly regulates the cell surface localization of MMP-7 through its interaction with pro-domain, xCelligence E-plate was coated with recombinant His-tagged pro-domain of MMP-7 (PDMMP-7) in the presence of either 10 μg of gelatin or 0.1 % BSA, and then cell adhesion of HT29-SDC2 cells seeded on the plates were analyzed (Figure 1C). As expected, PDMMP-7 on the plate enhanced cell attachment on the E-plates in both cases (Figure 1C), supporting the idea that MMP-7 directly interacts with cell surface SDC2 through the pro-domain.

### 2.2. Synthetic Peptide Derived from Syndecan-2 Inhibited Its Interaction with the MMP-7 Pro-Domain

It has been shown that the synthetic peptides (S2-P) corresponding to human SDC2 sequence bound to pro-domain of MMP-7 inhibits SDC2-mediated cancer activities [11], suggesting that S2-P could have potential as an anticancer peptide. However, S2-P had a relatively weak binding affinity for the MMP-7 pro-domain. Having found that Tyr 51 of S2-P was critical for the interaction [11], substitution of neighboring amino acid residues was applied in an effort to improve this interaction and possibly the anticancer activities (Figure 2). In addition to Tyr 51, assessment of the electrostatic plus shape interaction scoring suggested that the S2-P peptide interacted with the pro-domain of MMP-7 through Asp 49 of S2-P, which is located in a cluster of acidic amino acid residues at the N-terminal side of Tyr 51 [11]. We thus hypothesized that the acidic residues near Tyr 51 might influence the interaction affinity. Therefore, we either added acidic amino acids or changed acidic amino acids to nonpolar amino acids near Tyr 51. We generated four modified peptides from the original sequence of S2-P (Figure 2A), on the basis of stabilizing energy score by the Single Amino Acid Mutation Change of Binding Energy (SAAMBE) method [29]. The acidic amino acid residue was added at the C-terminus of Tyr residue (S2-D, S2-FE, and S2-FLD) or the acidic amino acid at the N-terminus of Tyr 51 residue was replaced by non-polar amino acid (S2-L, S2-FE, and S2-FLD) to increase their interaction with the MMP-7 pro-domain (Figure 2A). The ability of these peptides to physically interact with the MMP-7 pro-domain was subsequently evaluated using florescence spectrophotometer with purified recombinant SDC2 and a PDMMP-7 (Figure 2B and Appendix A). Compared to S2-P (Kd of 1.22 mM), S2-D, S2-FE, and S2-FLD showed greater binding activity to the pro-domain of MMP-7 (Kd of 0.55, 0.456, and 0.064 mM, respectively) (Figure 2B). In contrast, the Asp-to-Leu replacement (S2-L) reduced binding. As the addition of an acidic amino acid at the C-terminal side of Tyr 51 improved the interaction with the MMP-7 pro-domain, these data suggest that this interaction is favored by the maintenance of acidic surroundings around Tyr 51.

Since the synthetic peptides (e.g., S2-D, S2-FE, and S2-FLD) showed improved interaction with the MMP-7 pro-domain, we further examined whether these peptides competitively reduced the interaction of SDC2 with the MMP-7 pro-domain (Figure 3). We first examined whether the modified peptides could inhibit the attachment of HT29-SDC2 cells to E-plates coated with the pro-domain (Figure 3A). Among the four peptides, only S2-D and S2-FE reduced the attachment of HT29-SDC2 cells to the MMP-7 pro-domain to a degree stronger than that exhibited by S2-P (Figure 3A). Thus, both S2-D and S2-FE appeared to have enhanced ability to block the interaction of SDC2 with MMP-7 pro-domain at the surface of HT29-SDC2 cells. Consistently, S2-D and S2-FE decreased the cell-surface level of MMP-7 (Figure 3B) at the HT29-SDC2 cell surface in cells with similar expression levels of SDC2 and MMP-7 (Figure 3C and Appendix A). Together, these data suggest that the modified peptides, S2-D and S2-FE, efficiently inhibited the SDC2-mediated cell surface localization of MMP-7.

### 2.3. S2-D and S2-FE Show Anticancer Ability against Colon Cancer Cells

Since the interaction of SDC2 with MMP-7 is important to regulate cancer activity [11,26], and 50 nM of S2P showed anticancer activity both in vitro and in vivo [30], we further investigated whether 50 nM of S2-D and S2-FE could inhibit the cancer activity of HT29-SDC2 cells (Figure 4). Neither peptide showed cytotoxicity against HT29-SDC2 cells when applied at 50 nM for up to 48 h (Figure 4A). However, both peptides significantly reduced the SDC2-mediated inductions of migration (Figure 4B), invasion through Matrigel (Figure 4C), and colony formation on soft agar (Figure 4D). Unexpectedly, despite the improved interaction with the MMP-7 pro-domain, both SP-D and SP-FE showed inhibitory effects similar to that of S2-P in HT29-SDC2 cells (Figure 4).

To further investigate the anticancer activity of S2-D and S2-FE, HCT116 human colon cancer cells, which are more invasive cancer cells than HT29, were treated with 50 nM of each peptide and monitored changes in various cancer activities (Figure 5). As expected, the peptides did not exert cytotoxicity (Figure 5A) or influence the proliferation of HCT116 cells (Figure 5B). However, treatment of HCT116 cells with either S2-D or S2-FE decreased migration (Figure 5C), invasion through Matrigel (Figure 5D), and soft agar colony formation (Figure 5E). While the inhibitory activities of S2-D and S2-FE on cell migration were similar to that of S2-P (Figure 5C), S2-D showed inhibitory activities against invasion, and colony formation comparable to those of S2-P and S2-FE inhibited these parameters to an even greater degree in HCT116 cells (Figure 5D,E). Consistently, treatment of HCT116 cells with either S2-D or S2-FE decreased their cell surface co-localization of SDC2 with MMP-7 (Figure 5F). The values of Mander’s overlap coefficient (MOC), which measures the degree of overlap between SDC2 and MMP-7, were ≈0.838 in untreated control cells, ≈0.834 in S2-P-treated cells, ≈0.815 in S2-D-treated cells, and ≈0.803 in S2-FE-treated cells. Since lower MOC values indicate less overlap between SDC2 and MMP-7, the results suggest that S2-FE has better inhibitory effects on SDC2 and MMP-7 colocalization than S2-P on the cell surface of HCT116 cells. Together, these data suggest that S2-FE has better anticancer activity than S2-P.

### 2.4. S2-D and S2-FE Inhibited MMP-7-Mediated ECM Degradation in Colon Cancer Cells

Since the interaction with SDC2 induces the processing of pro-MMP-7 into its active form [26], interrupting this interaction with a synthetic peptide would be expected to reduce MMP-7 activity. Therefore, we further investigated the effect of S2-FE on the MMP-7-mediated ECM degradation of HCT116 cells (Figure 6). HCT116 cells were seeded on a plate coated with fluorescently labeled gelatin (a well-known substrate of MMP-7), and gelatin degradation was monitored under fluorescence microscopy. A total of 50 nM of the peptides were treated twice every 24 h. We did not detect any significant degradation of fluorescence-labeled gelatin by untreated HCT116 cells (Figure 6, top left), but HCT116 cells pretreated with interleukin 1α (IL-1α), which induces MMP-7 expression [31], were found to promote gelatin degradation. This degradation was significantly reduced by the application of S2-D or S2-FE (Figure 6, top right). These data suggest that S2-D and S2-FE can inhibit MMP-7 activation and thus MMP-7 activity at the cell surface of colon cancer cells.

### 2.5. The S2-FE Peptide Was Sufficient for Reducing Primary Tumor Growth and Metastasis

To further investigate anticancer activity of S2-D and S2-FE in vivo, a xenograft mouse model was employed (Figure 7 and Figure 8). To assess the effect of the peptides on the primary tumor growth of colon cancer cells, luciferase-expressing mouse colon cancer cells (CT26-luc) were injected subcutaneously into 6-week-old male BALB/c nude mice, and tumor growth was monitored up to 21 days after the injection of CT26-luc cells in either the absence or presence of the peptides (Figure 7). Subcutaneous images showed a significant decrease (14.03% reduction for S2-D, 64.58% reduction for S2-FE at day 21) in photon emission from the tumor sites in mice injected with either S2-D or S2-FE compared to S2-P-treated control mice (Figure 7A). S2-FE-treated mice exhibited slower tumor growth than S2-D-treated mice (Figure 7A). Correspondingly, the average tumor weight of S2-FE-treated mice was much less than that of S2-P-treated mice (Figure 7B).

To further evaluate the effects of the synthetic peptide in a metastasis model, we injected CT26-luc cells pre-incubated with the peptide into mouse tail veins. Analysis of lung samples revealed that both S2-D and S2-FE inhibited the pulmonary metastasis of CT-26 cells, but S2-FE showed a significant inhibitory effect (Figure 8A). Similar to the subcutaneous model, mice given S2-FE exhibited significantly slower tumor growth than S2-P-treated mice (Figure 8B). Together, these in vivo results suggest that S2-D and S2-FE have cancer-inhibiting effects, and that S2-FE exhibits more effective inhibition of primary tumor growth and metastasis than S2-P.

## 3. Discussion

We previously reported that the extracellular domain of SDC2 interacts with the MMP-7 pro-domain [32], and that a synthetic peptide corresponding to a portion of the human SDC2 sequence (S2-P) bound to the pro-domain of MMP-7 and inhibited SDC2-mediated cancer activities [11]. In this study, we generated modified peptides (S2-D, S2-L, S2-FE, and S2-FLD) from the original sequence of S2-P to improve the anticancer activity of S2-P and then investigated the anticancer activities of the peptides. Our data showed that, among four mimetic peptides, S2-D and S2-FE showed the higher binding affinity to the MMP-7 pro-domain (Figure 2B) and greater inhibitory effects on the SDC2–MMP-7 pro-domain interaction (Figure 2B) compared to S2-P, suggesting that S2-D and S2-FE could replace the endogenous MMP-7 interaction with cell surface SDC2, a cell surface docking receptor of MMP-7 [6,26]. Indeed, both S2-D and S2-FE decreased cell surface localization of MMP-7 (Figure 3B), which supports the ability of these peptides to prevent pro-MMP-7 from docking with SDC2. Given that MMP-7 activity critically regulates the cancer activity of colon cancer cells [11,32], it is highly possible that both S2-D and S2-FE could act as anticancer agents. As expected, S2-D and S2-FE showed a cancer-inhibiting effect against human colon cancer cells, as evidenced by the abilities of these peptides to reduce the migration, invasion, and soft-agar colony formation activity of HT29-SDC2 and HCT116 cells (Figure 4 and Figure 5). Since the interaction of SDC2 with MMP-7 induced the activation of MMP-7 [11], we hypothesized that disruption of their interaction could decrease MMP-7 activity. Indeed, S2-D and S2-FE reduced the MMP-7-mediated degradation of gelatin substrate (Figure 6). However, while S2-D showed similar inhibitory activity compared to S2-P in all cancer activities tested, S2-FE showed an improved anticancer activity in comparison with S2-P against HCT116 cells (Figure 5). In addition, S2-FE, but not S2-D, showed better inhibitory effect on primary tumor growth (Figure 7) and lung metastasis (Figure 8) of CT26 cells than S2-P peptide. Together, these data suggest that modifying S2-P by replacing an acidic amino acid in the vicinity of Tyr 51 enhanced the interaction with pro-MMP-7 and thereby inhibited MMP-7 activity and the cancer activities of colon cancer cells. These findings further suggest that S2-FE could be effective anticancer peptides against colon cancer.

Many efforts have been made to develop anticancer therapeutics targeting MMPs, with a particular focus on their proteolytic activities. The developed agents include hydroxamate-based chelation inhibitors (e.g., batimastat [33], marimatat [34], prinomastat [35]). Although such agents have exhibited promising antitumor effects in animal models of various cancers, the clinical trials have been largely unsuccessful due to severe side effects [36]. For instance, batimastat showed malignant pleural effusion and ascites in a phase I study [37,38] and marimastat was canceled in phase III clinical trials because it had significant musculoskeletal toxicity [35,39]. Since the various MMPs are widely distributed throughout the body, structurally conserved, and involved in many different aspects of cell functions, agents that block MMP protease activity can have huge impacts on cell functions. This may limit the usefulness of strategies directly targeting the enzymatic domain of MMPs. Interestingly, this study implies that MMP-7 activity can be specifically regulated through a direct interaction with SDC2 on the cancer cell surface, and that blocking this activating interaction could be a means to specifically inhibit the enzymatic activity of MMP-7.

MMPs are secreted as catalytically inactive pro-MMPs [12]. To obtain the enzymatic activity of MMPs, cleavage of pro-domain is necessary. It is well known that the docking of pro-MMP-2 with TIMP-2 facilitates the activation of MMP-2 [25]. This interaction elevates the density of pro-MMP-2, and MT1-MMP forms a complex with pro-MMP-2 and TIMP-2 to enable the activating cleavage of the MMP-2 pro-domain [25,40]. To inhibit the activity of MMP-2 in human melanoma cells, researchers previously used a cyclic peptide to target the protein–protein interaction of pro-MMP-2 and TIMP-2 [41]. Although a previous study showed that the docking of pro-MMP-7 to SDC2 elevated the activation of MMP-7 [11], the exact underlying mechanism remained unknown. Similar to the case of MMP-2, the interaction of SDC2 with pro-MMP-7 may cause molecules of the latter to cluster via SDC2-mediated homodimerization [42], and the clustered pro-MMP-7 molecules may cleave one another’s pro-domains in a cross-cutting manner. Further studies are needed to clarify the mechanism underlying the SDC2-prompted activating cleavage of pro-MMP-7 at the colon cancer cell surface.

Indeed, the abilities of S2-FE to inhibit MMP-7 activity and the tumorigenic activities of colon cancer cells suggest that these peptides could potentially serve as effective anticancer drugs for colon cancer. However, since therapeutic peptides composed of natural amino acid sequences are known to be easily degraded, relatively unstable, and sensitive to proteases of the digestive system and blood plasma [43], further studies will be required to improve the stability of the peptides. Possible strategies for this include cyclization, bioisosteric replacement of peptide bonds, and changing the stereochemistry of amino acids [44].

In summary, we herein reveal that mimetic peptides derived from SDC2 (e.g., S2-FE) can interrupt the interaction of SDC2 with pro-MMP-7 and thus suppress colon cancer cell activities. Since S2-FE have improved anticancer activity over the parent peptide S2-P, S2-FE could be promising anticancer peptides against colon cancer. Although future studies will be required to clarify the underlying inhibitory mechanism and improve the stability of these peptides before they can be developed as new anticancer drugs, the present study lays new groundwork for the specific targeting of MMP-7 via SDC2 in colon cancer.

## 4. Materials and Methods

### 4.1. Materials

A monoclonal antibody capable of recognizing human, rat, and mouse SDC2 was produced by AdipoGen Inc. (Incheon, Korea) using the Fc-fused extracellular domain of SDC2 [28]. A polyclonal antibody that recognizes human, rat, and mouse SDC2 was produced by AbClon (Seoul, Korea) using a human SDC2 extracellular domain peptide. Monoclonal anti-MMP-7 was purchased from R&D Systems (Minneapolis, MN, USA). The quenched fluorescence MMP-7 substrate, (7-methoxycoumarin-4-yl) acetyl-Pro-Leu-Gly-LeuN-3(2,4-dinitrophenyl)-L-2,3 diaminopropionyl Ala-Arg-NH_2_, was purchased from Bachem (Bubendorf, Switzerland). Human recombinant MMP-7 proenzyme was purchased from Sigma-Aldrich (St. Louis, MO, USA). Oregon-Green-488-conjugated gelatin from pig skin was purchased from Invitrogen (Waltham, MA, USA).

### 4.2. Cell Culture

The human colon adenocarcinoma cell lines HT29 and HCT116 were purchased from the American Type Culture Collection (ATCC, Manassas, VA, USA) and maintained in McCoy’s 5A complete medium (Welgene, Daegu, Korea) supplemented with 10% (*v/v*) fetal bovine serum (FBS; Hyclone, Logan, UT, USA) and gentamycin (50 g/mL: Sigma-Aldrich). Moreover, CT26 cell line stably transfected with firefly luciferase (CT26-Luc), a colon cancer cell line generated from BABL/C mice, was purchased from the American Type Culture Collection (ATCC, Manassas, VA, USA) and cultured in RPMI-1640 medium (Hyclone) supplemented with 10% FBS and gentamicin (50 μg/mL). All cell lines were incubated at 37 °C in a 5% CO_2_-containing humidified atmosphere.

### 4.3. Vector Construction and Generation of Stable Cell Lines

To generate cell lines stably expressing SDC2, HT29 cells (1 × 10^5^) were transfected with 1 μg of empty pcDNA3.1 (control) or pcDNA3.1-FLAG-SDC2 (wild-type; Appendix A) [28] and selected in medium containing 400 μg/mL G418 (EMD Biosciences, San Diego, CA, USA) for 4 weeks. The surviving clones were individually isolated and analyzed by FACS and RT-PCR.

### 4.4. Peptide Synthesis

The peptide corresponding to human SDC2 extracellular domain residues 41–60 (S2-P: SGVYPIDDDDYASASGSGAD) and its modified peptides (S2-D; SGVYPIDDDDYDSASGSGAD, S2-L; SGVYPIDDLDYASASGSGAD, S2-FE; SGVYPIDDFDYESASGSGAD, S2-FLD; SGVYPIFDLDYDSASGSGAD) were synthesized using an improved version of the Fmoc chemistry-based solid-phase method (Anygen Inc., Kwangju, Korea).

### 4.5. Expression and Purification of Recombinant Pro-Domain of MMP-7

The gene encoding the pro-domain sequence (PDMMP-7) was cloned into the pET32a vector, which has a TRX-His tag at the N-terminus, for protein expression and purification. Between the TRX-His tag and the pro-domain, a TEV cleavage sequence was inserted. Proteins were overexpressed with 1 mM IPTG and isolated using Ni-NTA affinity chromatography. The purified TRX-His tag-PDMMP-7 was incubated with TEV protease at a 1:1 molar ratio for 12 h. The reaction mixture was further purified in phosphate-buffered saline (PBS) buffer at pH 7.4 using a HiLoad^TM^ superdex^TM^ 75 column (GE Healthcare, Little Chalfont, UK) to remove tags. The final protein sample was concentrated to 14 mg/mL for the fluorescence experiment.

### 4.6. Fluorescence Assay

The binding affinity between the PDMMP-7 and each synthetic peptide was measured at 298 K (25 °C) using an LS55 fluorescence spectrophotometer (Perkin Elmer, Waltham, MA, USA) at wavelengths of 280 nm (excitation) and 300–450 nm (emission). PDMMP-7 and peptides were prepared in phosphate-buffered saline (PBS) buffer at pH 7.4. The concentration of PDMMP-7 was set at 10 μM, and the input amount of each peptide versus PDMMP-7 was titrated at a molar ratio up to 1:80 using a thermostat cuvette. The dissociation constant (Kd) of the PDMMP-7/peptide complex was calculated using the equation log[(Fo − F)/F] = log(1/Kd) + nlog[ligand], where Fo and F represent the fluorescence intensity of PDMMP-7 at 347 nm in the absence and presence of the peptide, respectively, and n represents the number of binding sites.

### 4.7. MMP Enzyme Activity Assay

The catalytic activity of MMP was analyzed by a peptide cleavage assay using the quenched fluorescent substrate 7-methoxycoumarin-4-yl acetyl-Pro-Leu-Gly-LeuN-3(2,4-dinitrophenyl)-L-2,3 diaminopropionylAla-Arg-NH_2_ for MMP-7 and 7-methoxycoumarin-4-yl-Pro-Leu-Ala-L-norvaline-L-2,3 diaminopropionylAla-Arg-NH_2_ (a generous gift from Dr. Dong Hae Shin at Ewha Womans University) for MMP-2. Conditioned media (CM) were collected and concentrated using 10 K Amicon^®^ Ultra Centrifugal Filter Units (Merck Millipore, Burlington, MA, USA). The reactions were performed in a final volume of 200 μL of MMP assay buffer (20 mM Tris HCl (pH 7.4), 150 mM NaCl, 5 mM CaCl_2_, 0.5 mM ZnCl_2_, 0.001% Brij35) in the presence of 1 μM oligopeptide and 50 ng of human recombinant MMP-7 proenzyme (Sigma-Aldrich, St. Louis, MO, USA) or 50 μL of CM for 1.5 h at 37 °C. Fluorescence was determined at an excitation wavelength of 328 nm and an emission wavelength of 395 nm using a SpectraMax^®^ i3 plate reader (Molecular Devices, Sunnyvale, CA, USA) [28].

### 4.8. Monitoring of Cell Attachment

Cell adhesion was monitored in real time using an xCELLigence system (Roche Diagnostics GmbH, Basel, Switzerland). The bottom of an E16 xCelligence plate (ACEA Biosciences, Santa Clara, CA, USA) was coated with a mixture of gelatin and PDMMP-7 at 37 °C for 1 h. The plate was washed with PBS, loaded with serum-free medium (50 μL/well), and incubated at 37 °C in 5% CO_2_ for 15 min. The background was measured using an RTCA DP Analyzer (RTCA software version 1.2, ACEA Biosciences). HT29-SDC2 cells (2 × 10^4^ cells/well) were added to each well with SDC2 mimetic peptides (50 nM), and the plate was incubated at 37 °C in 5% CO_2_ for 15 min. After 15 min, the plate was assembled onto the RTCA DP Analyzer, and cell adhesion was assessed at 2-minute intervals for 1 h at 37 °C under 5% CO_2_. The data obtained were analyzed using the provided RTCA software.

### 4.9. Immunofluorescence Analysis

Cells were seeded onto glass coverslips in 12-well plates, incubated for 48 h (HT29 3 × 10^5^ cells/well, HCT116 2 × 10^5^ cells/well), and treated with 50 nM of SDC2 peptide. After 24 h from peptide treatment, cells were fixed with 3.5% paraformaldehyde in PBS at room temperature for 10 min. The cells were rinsed three times with PBS, blocked with 0.5% bovine serum albumin (BSA) in PBS for 1 h, washed, and stained with the appropriate primary antibody overnight at 4 °C. The cells were then washed with PBS and incubated with FITC-conjugated mouse antibodies (Thermo Fisher Scientific, Waltham, MA, USA) and Texas-Red-conjugated rabbit antibodies (Thermo Fisher Scientific) for 1 h at room temperature. The coverslips were washed with PBS and mounted on glass slides with mounting solution containing 4′,6-diamidino-2-phenylindole (DAPI), and the results were imaged under a confocal fluorescence microscope (Carl Zeiss, Gottingen, Germany) [11].

### 4.10. RT-PCR

Total RNA was isolated from cells using an easy-BLUE kit (Intron, Seoul, South Korea). The RNA was extracted with chloroform and precipitated with isopropanol. The RNA pellet was washed with 75% ethanol and resuspended in DEPC-treated water. Approximately 3 μg of RNA was used to generate cDNA using AMV Reverse Transcriptase (Cat# M5108) and Random Primer (Cat# C1181) (both from Promega US, Madison, WI, USA). Aliquots of the resulting cDNAs were amplified using following primers: rat syndecan-2, 5′-ATGCGGGTACGAGCCACGTC-3′ (forward) and 5′-CGGGAGCAGCACTAGTGAGG-3′ (reverse); human SDC2 5′-ACATCTCCCCTTTGCTAACGGC-3′ (forward) and 5′-TAACTCCATCTCCTTCCCCAGG-3′ (reverse), human MMP-7 5′-GGTCACCTACAGGATCGTATCATAT-3′ (forward) and 5′-CATCACTGCATTAGGATCAGAGGAA-3′ (reverse), human GAPDH 5′-CCACCCATGGCAAATTCCATGGCA-3′ (forward) and 5′-TCTAGACGGCAGGTCAGGTCCACC-3′ (reverse). After an initial denaturation at 94 °C for 5 min, the samples were subjected to 30 cycles of denaturation at 94 °C for 30 s, annealing at 55 °C for 60 s, and extension at 72 °C for 60 s. Human GAPDH was amplified as an internal control. The generated PCR products were separated by 1% agarose gel electrophoresis.

### 4.11. Cell Proliferation Assay

Cell proliferation was measured by a colorimetric assay using 3-(4,5-dimethythiazol-2-yl) 2,5-diphenyltetrazolium bromide (MTT; Amresco, Solon, OH, USA) according to the manufacturer’s instructions. Briefly, cells were harvested with 0.05% trypsin/EDTA and seeded to 48-well plates (Thermo Fisher Scientific). After cells were allowed to attach to the plate for 24 h, medium containing 0.5 mg/mL MTT was added to each plate and incubation was continued for 1 h. The medium was then removed, and 200 μL of dimethyl sulfoxide (DMSO) was added to each plate for 30 min at room temperature. The mean concentration of absorbance at 570 nm in each set of samples was measured using a 96-well microtiter plate reader (Dynatech, Chantilly, VA, USA).

### 4.12. Cell Cycle Analysis

Cells were seeded to 6-well plates (HT29, 3 × 10^5^ cells/well; HCT116, 2 × 10^5^ cells/well) and incubated for 48 h. Peptides were applied for 24 h. The cells were then washed with PBS, released, and fixed with 5 mL of cold 70% ethanol in PBS at 4 °C. The ethanol was discarded, and the cells were washed with cold PBS and incubated with 0.1 mg/mL of propidium iodide (Sigma-Aldrich) and 0.6% Triton-X 100 in PBS with RNase at 25 °C for 45 min. Cell cycle parameters were analyzed by flow cytometry.

### 4.13. Invasion and Migration Assay

For the migration assay, gelatin (10 μg/mL) was added to each well of a Transwell plate (8 μm pore size; Costar, Corning, NY, USA), and the membranes were allowed to dry at 25 °C for 1 h. The Transwell plates were assembled into a 24-well plate, and the lower chamber was filled with McCoy’s 5A medium containing 10% FBS, 1% BSA, and basic fibroblast growth factor (10 μg/mL). Cells (HT29, 1 × 10^6^ cells/well; HCT116, 5 × 10^5^ cells/well) were added to each upper chamber, and the plate was incubated at 37 °C in a 5% CO_2_ incubator. The cells that had migrated to the lower surface of the filters were stained with hematoxylin and 0.5% eosin and counted. For the in vitro invasion assay, Transwell plates were coated with gelatin (10 μg/mL) on the lower side of the membrane and with Matrigel (BD Biosciences, Mississauga, Canada) (3 mg/mL) on the upper side of the membrane.

### 4.14. Anchorage-Independent Growth in Soft Agarose

Each well of a 6-well culture plate was coated with 3 mL of bottom agar mixture (McCoy’s 5A containing 10% FBS and 0.6% agar). After the bottom layer had solidified, 1 mL of top agar mixture (McCoy’s 5A containing 10% FBS and 0.3% agar) containing cells (HT29, 2 × 10^5^ cells/well; HCT116, 1 × 10^5^ cells/well) was added to each well, and the cultures were incubated at 37 °C in a 5% CO_2_ atmosphere. Colony formation was monitored daily with a light microscope. After 14 days, the colonies were stained with 0.005% crystal violet and photographed with a digital camera.

### 4.15. Gelatin Degradation Assay

Oregon-Green-488-conjugated gelatin from pig skin was dissolved at 1 mg/mL in 2% sucrose. Coverslips in 24-well plates were coated with 0.2 mg/mL Oregon-Green-488-conjugated gelatin at 4 °C for 20 min in the presence of 40 μL of 0.5% glutaraldehyde diluted in 1 mL PBS. The coverslips were washed three times in PBS at room temperature and incubated with 5 mg/mL NaBH_4_ (dissolved in PBS) at room temperature for 3 min. The coverslips were washed as above, sterilized in 70% ethanol for 1 min, and then dried. After being incubating in McCoy’s 5A medium at 37 °C for 1 h, 1.25 × 10^5^ HCT116 cells mixed with or without 1 ng/mL of IL-1α and 50 nM of SDC2 peptide were seeded onto coverslips in 24-well plates (Costar, Corning, NY, USA) and cultured for 20–72 h at 37 °C in a 5% CO_2_ atmosphere. After 24 h, 50 nM of SDC2 peptide were added in the treated 24-well plates. The cells were fixed with 3.5% paraformaldehyde for 10 min and washed, and the coverslips were mounted on microscopic slides and imaged by fluorescent microscopy.

### 4.16. Mouse Model

For the subcutaneous model, CT26-luc cells (5 × 10^6^ cells/mouse) incubated with synthetic peptide (final 250 nM) at 37 °C for 30 min in 100 μL of PBS were subcutaneously injected below the dorsal flank into 6-week-old male BALB/c mice (*n* = 5 per group). For imaging, mice were intraperitoneally injected with 150 mg/kg D-luciferin and anesthetized with 1% isoflurane. At 10–20 min after D-luciferin injection, mice were placed in an IVIS Imaging System (IVIS SPECTRUM; Caliper Life Sciences, Waltham, MA, USA) and imaged dorsally. Tumor growth was monitored weekly by the IVIS Imaging System, and external caliper measurements were taken (L × W × D) for 21 days. For the in vivo experimental pulmonary metastasis assay, CT26-luc cells (1 × 10^5^ cells/mouse) incubated with synthetic SDC2 peptide (final 250 nM) at 37 °C for 30 min in 200 µL of PBS were injected via the tail veins into 6-week-old male BALB/c mice (*n* = 3 per group; Orient Bio Co., Seoul, Korea). Growth of metastatic lung tumors was monitored weekly by the IVIS Imaging System; images were captured on days 6, 16, and 19. On day 19, mice were sacrificed, lungs were excised, and metastatic nodules were photographed and counted. Animal study was conducted in accordance with the Institutional Animal Care and Use Committee (IACUC) of the National Cancer Center Research Institute (NCCRI), and the IACUC approval number is NCC-21-680 (the date of approval 4 August 2021)

### 4.17. Statistical Analysis

All data are presented as mean ± S.D. Differences between groups were tested for statistical significance using Student’s *t*-test and were considered significant at * *p* < 0.05 or ** *p* < 0.01.

## Figures and Tables

**Figure 1 ijms-23-05888-f001:**
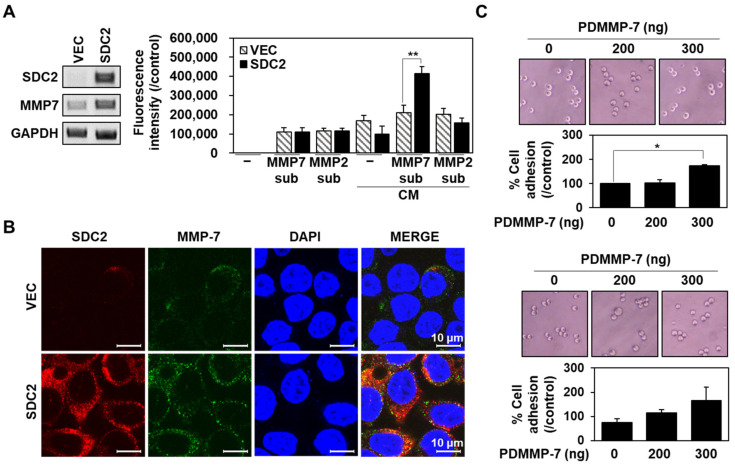
Syndecan-2 mediates the localization of pro-MMP-7 by interacting with the pro-domain of MMP-7. (**A**) HT29 cells were stably transfected with empty vector (HT29-VEC) or a vector encoding syndecan-2 (HT29-SDC2), and the mRNA expression level of sydecan-2 (SDC2) was analyzed by RT-PCR (left). Proteolytic activities of MMP-7 and MMP-2 in the conditioned media (CM) were evaluated by a quenched fluorescence peptide cleavage assay performed using specific substrates (Mca-Pro-Leu-Gly-Leu-Dap(Dnp)-Ala-Arg-NH_2_ for MMP-7 and Mca-PLA-Nva-Dap(Dnp)-AR-NH_2_ for MMP-2) (right). Fluorescence data were normalized by subtracting values from the control without substrate. Data are shown as mean ± S.D. (*n* = 3); **, *p* < 0.01 versus MMP-7 activity of HT29-VEC. (**B**) Cells were incubated with anti-SDC2 antibody and stained with a Texas-Red-conjugated secondary antibody or incubated with anti-MMP-7 antibody and stained with a FITC-conjugated secondary antibody. Photographs were obtained under confocal microscopy. Scale bar: 10 μm. (**C**) HT29-SDC2 cells were plated on E-plates pre-coated with gelatin 10 μg (top) or 0.1% BSA (bottom) with the indicated amounts of His-tagged MMP-7 pro-domain (PDMMP-7). After 4 h, cells were photographed under phase-contrast microscopy, and cell attachment was monitored and plotted using the xCELLigence system. The mean of cell attachment ± S.D. is shown (*n* = 3); *, *p* < 0.05 versus 0 ng of PDMMP-7.

**Figure 2 ijms-23-05888-f002:**
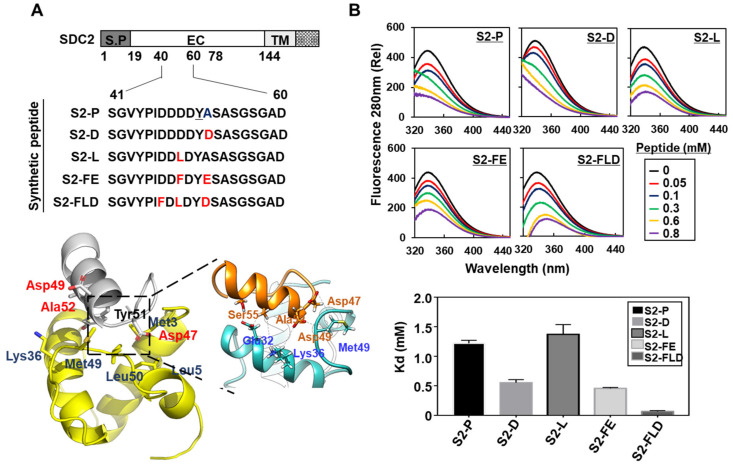
Synthetic peptides derived from SDC2 interacted with the MMP-7 pro-domain. (**A**) Schematic representation of the SDC2 core protein. The signal peptide (SP), the extracellular domain (EC), the transmembrane domain (TM), and the cytoplasmic domain (CT) of SDC2 are noted. Amino acid sequences of the peptide corresponding to residues 41–60 of the SDC2 extracellular domain (S2-P) and its modified peptides (S2-D, S2-L, S2-FE, S2-FLD) are shown. Tyr 51, which is critical for the interaction, is underlined, and alteration sites of the amino acid sequence are colored red. The docking structure of S2-P and PDMMP-7 is visualized using the ribbon diagram. (**B**) Interaction of PDMMP-7 with the indicated peptides was analyzed with fluorescence spectroscopy (top). Titration between PDMMP-7 and the peptides was performed, and the Kd value was calculated (bottom).

**Figure 3 ijms-23-05888-f003:**
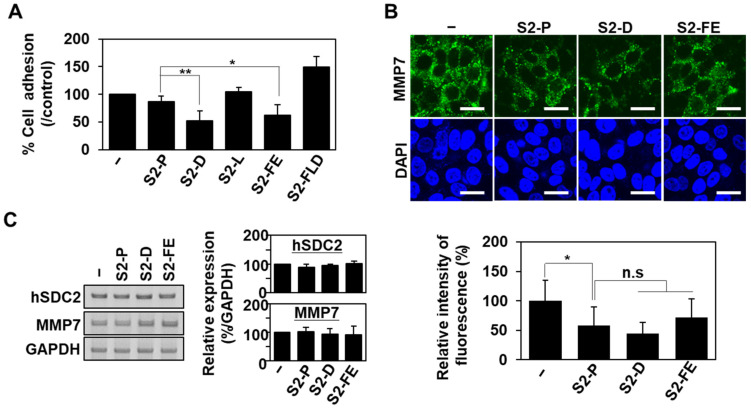
The synthetic peptides inhibited the interaction of SDC2 with the MMP-7 pro-domain. (**A**) HT29-SDC2 cells were plated on E-plates pre-coated with 10 μg gelatin and 300 ng of PDMMP-7 in the presence of 50 nM of the indicated peptides. Cell attachment was monitored for 1 h and plotted using the RTCA software. Data are shown as mean ± S.D. (*n* = 5); *, *p* < 0.05, **, *p* < 0.01 versus S2-P. (**B**) HT29-SDC2 cells pretreated with 50 nM of the indicated peptides for 24 h were incubated with anti-MMP-7 antibody and stained with an FITC-conjugated secondary antibody. Photographs were obtained under confocal microscopy. Scale bar: 20 μm. Quantitative analysis of fluorescence intensity were performed using ImageJ program. Data are shown as mean ± S.D. (*n* = 6); *, *p* < 0.05; n.s, no significant versus control or S2-P. (**C**) HT29-SDC2 cells were incubated with 50 nM of each peptide for 24 h, and the expression levels of the target mRNAs were analyzed by RT-PCR.

**Figure 4 ijms-23-05888-f004:**
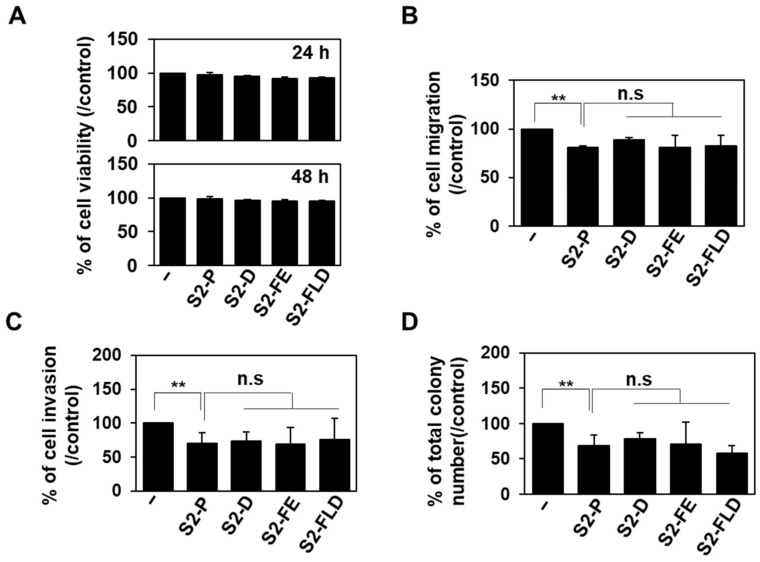
The synthetic peptides inhibited the tumorigenic activity of HT29-SDC2 cells. (**A**) The proliferation of HT29-SDC2 cells was evaluated with the MTT assay as described in the ‘Materials and Methods’ section. Data are shown as the average value ± S.D. of three independent experiments carried out in triplicate. (**B**) Transwell migration assays were performed with HT29-SDC2 cells treated with 50 nM of the peptide, with 10% FBS used as a chemoattractant in the lower chamber. Cells (1 × 10^6^) were allowed to migrate on gelatin-coated (10 μg/mL) transwell plates for 24 h. Data are shown as mean ± S.D. (*n* = 3); n.s, no significant; **, *p* < 0.01 versus HT29-SDC2 control cells or S2-P treated HT29-SDC2 cells. (**C**) For the invasion assay, cells were loaded to the upper compartments of Matrigel-coated (30 μg/mL) plates, incubated for 24 h, and fixed and stained with 0.6% hematoxylin and 0.5% eosin. The number of migrated or invasive cells was counted. Data are shown as mean ± S.D. (*n* = 3); n.s, no significant; **, *p* < 0.01 versus HT29-SDC2 control cells or S2-P treated HT29-SDC2 cells. (**D**) HT29-SDC2 cells (2 × 10^5^/dish) were seeded in soft-agar plates and allowed to grow for 14 days, and the viable colonies were counted. Data are shown as mean ± S.D. (*n* = 3); n.s, no significant; **, *p* < 0.01 versus HT29-SDC2 control or S2-P treated HT29-SDC2 cells.

**Figure 5 ijms-23-05888-f005:**
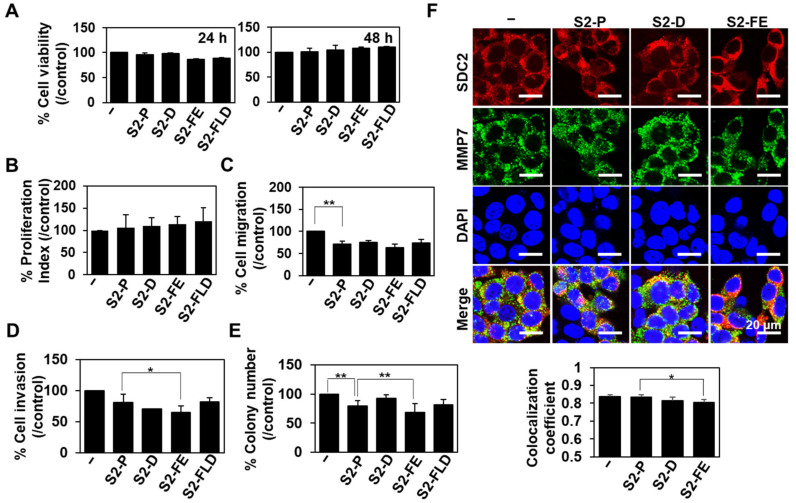
The synthetic peptides inhibited the tumorigenic activity of HCT116 cells. (**A**) The proliferation of HCT116 cells was evaluated with the MTT assay. Data are shown as the average value ± S.D. of three independent experiments carried out in triplicate. (**B**) Cell cycle profiles were obtained by propidium iodine staining and flow cytometry. The proliferation index was calculated as the sum of the number of cells in S phase and G2/M phase. Data are shown as mean ± S.D. (*n* = 3). (**C**) Transwell migration assays were performed with HCT116 cells treated with 50 nM of the peptide, with 10% FBS used as a chemoattractant in the lower chamber. Cells (5 × 10^5^) were allowed to migrate on gelatin-coated (10 μg/mL) transwell plates for 24 h. Data are shown as mean ± S.D. (*n* = 3); **, *p* < 0.01 versus control. (**D**) For the invasion assay, cells were loaded to the upper compartments of Matrigel-coated (30 μg/mL) plates, incubated for 24 h, and fixed and stained with 0.6% hematoxylin and 0.5% eosin. The migrated (invaded) cells were counted. Data are shown as mean ± S.D. (*n* = 3); *, *p* < 0.05 versus S2-P. (**E**) HCT116 cells (1 × 10^5^/dish) were seeded in soft-agar plates and allowed to grow for 14 days, and the viable colonies were counted. Data are shown as mean ± S.D. (*n* = 3); **, *p* < 0.01 versus control or S2-P. (**F**) HCT116 cells were treated with 50 nM of the indicated peptides for 24 h and incubated with anti-SDC2 antibody and stained with a Texas-Red-conjugated secondary antibody or incubated with anti-MMP-7-specific antibody and stained with an FITC-conjugated secondary antibody. Photographs were obtained under confocal microscopy. Scale bar: 20 μm. *, *p* < 0.05 versus S2-P.

**Figure 6 ijms-23-05888-f006:**
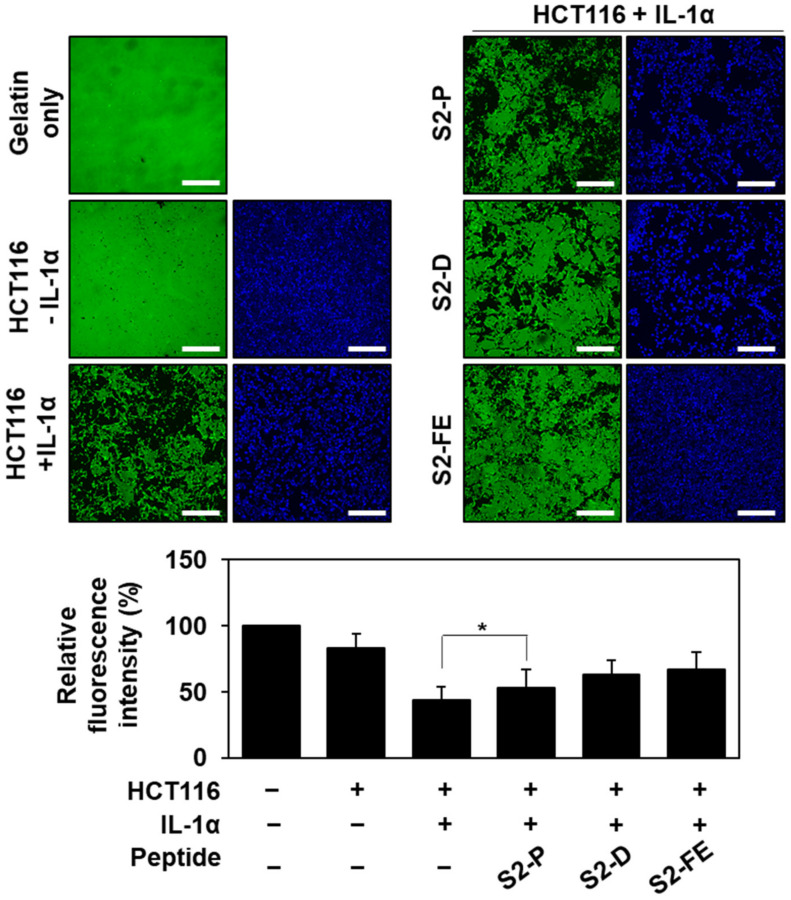
The synthetic peptides inhibited the activity of MMP-7 in colon cancer cells. HCT116 cells were treated with or without 1 ng/mL of interleukin 1 alpha (IL-1α) and plated on cover glasses in a 24-well plate precoated with fluorescent Oregon-Green-488-conjugated gelatin substrate in the presence or absence of the indicated peptides (50 nM). After 24 h, additional peptides were treated (50 nM). After total 48 h incubation, microscopic fields were photographed. Representative fluorescent images are shown. Scale bar: 200 μm. Quantitative analysis of fluorescence intensity of gelatin in cover glasses were performed using ImageJ program. Data are shown as mean ± S.D. (*n* = 3); *, *p* < 0.05 versus control.

**Figure 7 ijms-23-05888-f007:**
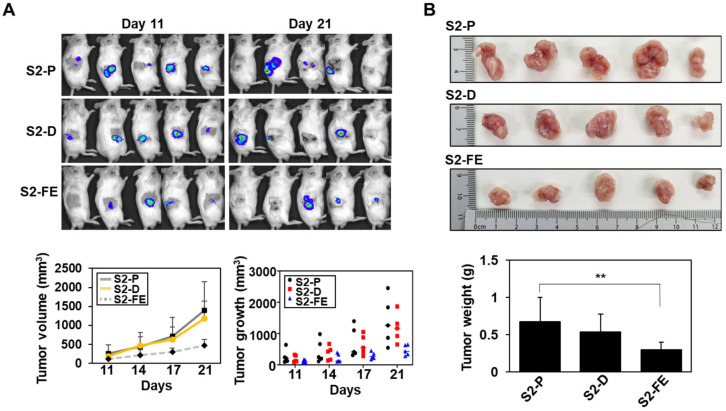
The S2-FE peptide inhibited subcutaneous tumor growth in vivo. (**A**) CT26-luc cells (5 × 10^6^ cells/mouse) pre-incubated with peptides (final 250 nM) were subcutaneously injected below the dorsal flank into 6-week-old male BALB/c mice (*n* = 5 per group). Representative images of in vivo tumor development at the injection sites are shown. Representative photographs of each primary tumor are shown (top). Average tumor volume (mm^3^) at 3–4 days post-injection is shown (bottom left). Data are presented as mean ± S.D. (*n* = 3). Tumor volume (mm^3^) at the indicated days post-injection for each individual mouse (*n* = 5) are shown (bottom right). (**B**) External caliper measurements were taken (L × W × D) for 21 days, and tumor weight (g) was assessed at the endpoint of the experiment. The columns represent the mean ± S.D. of tumor weight (*n* = 5). **, *p* < 0.01 versus S2-P.

**Figure 8 ijms-23-05888-f008:**
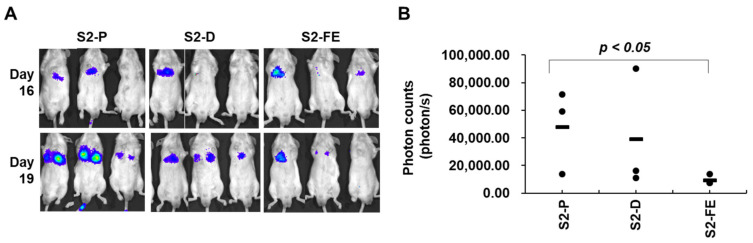
The S2-FE peptide decreased metastasis in mouse models. (**A**) BALB/c mice (*n* = 3 per group) were tail-vein-injected with CT26-luc cells (1 × 10^5^ cells/mouse) that had been pre-incubated with peptides (final 250 nM). Mice were sacrificed at 3 weeks post-injection. Representative images of in vivo tumor development are shown. Tumor growth was quantified (as photon/s) weekly by IVIS beginning at 6 days post-injection. (**B**) The tumor signal for 19 days was quantified from the photon count (*n* = 3), *p* < 0.05 versus S2-P.

## Data Availability

Available by contacting the corresponding author.

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
