# Peer review of "Substituted Syndecan-2-Derived Mimetic Peptides Show Improved Antitumor Activity over the Parent Syndecan-2-Derived Peptide"

_ijms, 2022, doi:10.3390/ijms23115888_

Round 1
Reviewer 1 Report
The manuscript authored by Bohee Jang et al. addresses the finding of novel substituted Syndecan-2-Derived peptides with potential improved therapeutic effect in cancer. Although the idea of finding such peptides deserves attention there are some points which should be addressed before considering a potential publication. For most of the cell culture-related results the authors test significance only between no peptide and any version of the peptide S2-P/S2-D or S2-FE. Instead they should test particularly between S2-D/S2-FE and the already published peptide S2-P to indeed demonstrate that the new peptides perform better compared with the original one. Also there is no justification as why S2-FLD is missing from most of the experiments and I find that there are some controls missing from their reported animal experiments. Below I have detailed the most important concerns:
Major points:
- Figure 1 and supplementary: Please add molecular weight ladder. What is the molecular weight expected for the SDC2 construct and MMP-7? Please indicate this in the text. What is the sequence homology between the rat and human version of SDC2? Please add in the supplementary section a figure with sequence alignment between human and rat SDC2.
- Why does the S2-FLD peptide which shows the lowest Kd (thus the greatest affinity) does not inhibit cell adhesion, but on the contrary it improves according to results in Figure 3A? This suggests that this interaction is not directly responsible for the differences in cell adhesion in Figure 3A. The authors should comment on this aspects in the manuscript.
- Figure 3B: It appears no difference between S2-P and any other peptide variant. Also why S2-FLD was not tested? Please add S2-FLD here and to the rest of the experiments through the manuscript.
- Figure 3C: Please add some parameter (Mander's overlap coefficient or Pearson correlation coefficient) to quantify the degree of colocalization. Also a statistical test should be used to assess the differences.
- p5, L179-182: The authors state: ‘These inhibitory activities of S2-D and S2-FE were stronger than those of S2-P …’, but the results show no changes or only modest changes compared to S2-P in Figures 3B and C. In my opinion the only real change is between no peptide and any other peptide tested (S2-P, S2-D, S2-FE). Please add real convincing results to sustain this claim or rephrase.
- p6 L202 and Figure 4: How do the authors choose the peptide concentration to test the cytotoxicity? How does this compare with the ones used for the results in Figure 3 or Figure 2? Please explain in the manuscript the rationale for this. Also, for the best performing peptide a gradient of concentrations could be tested to demonstrate the specificity of the effect.
- Figure 4B-D and through the rest of the manuscript: Again the authors should test statistically if it is any difference between S2-P and S2-D/S2FE not between no peptide and the rest of the conditions. Otherwise they cannot claim that S2-D and S2-FE compares better to S2-P (the WT).
- Figure 5C: How many cells were included for the MOC calculation? Please include a graph with values calculated for each cell analyzed and test if any significant difference can be found between S2-P and S2-D/S2-FE to verify if the two peptides indeed decrease the co-localization of SDC2 and MMP-7, compared to the initial peptide. Just declaring some values does not help to sustain the author’s conclusion.
- Figure 7 and 8: Please add similar images and measurements for the animals injected with CT26-luc cells not treated with any peptide as stated in the text. Also following that mice were sacrificed the author’s should demonstrate the presence of the peptides at the end of the experiment.
Minor points:
- p3L100-102: ‘In this study, we further determined whether those interaction is sufficient for their interaction…’. Please rephrase.
- Figure 1A: The bars represent mean raw fluorescence or normalized values? If yes to what is normalized? Please describe in the figure legend.
- Figure 1B: Please use a single notation through the manuscript either SDC-2 or SDC2.
- Figure 8: Missing p values and no peptide control.
Author Response
Dear Editors
We appreciate the thoughtful reviews of our manuscript entitled “Substituted syndecan-2-derived mimetic peptides show improved antitumor activity over the parent syndecan-2-derived peptide”.
We have carefully reviewed the reviewers’ comments. Since we have addressed all of the reviewers’ concerns and carefully revised our manuscript, we believe that we have improved the manuscript to the point that it should be suitable for publication. Therefore, we wish to submit the revised manuscript. Our responses to the reviewers’ concerns are explained point by point below.
This work is not under consideration by any other journal at this time, and the data have not been previously published. We appreciate your consideration and hope you will find this revised manuscript suitable for publication in IJMS.
Eok-Soo Oh
Reviewer 1
The manuscript authored by Bohee Jang et al. addresses the finding of novel substituted Syndecan-2-Derived peptides with potential improved therapeutic effect in cancer. Although the idea of finding such peptides deserves attention there are some points which should be addressed before considering a potential publication. For most of the cell culture-related results the authors test significance only between no peptide and any version of the peptide S2-P/S2-D or S2-FE. Instead they should test particularly between S2-D/S2-FE and the already published peptide S2-P to indeed demonstrate that the new peptides perform better compared with the original one. Also there is no justification as why S2-FLD is missing from most of the experiments and I find that there are some controls missing from their reported animal experiments. Below I have detailed the most important concerns:
Major points:
(1) Figure 1 and supplementary: Please add molecular weight ladder. What is the molecular weight expected for the SDC2 construct and MMP-7? Please indicate this in the text. What is the sequence homology between the rat and human version of SDC2? Please add in the supplementary section a figure with sequence alignment between human and rat SDC2.
Response: As suggested, we added a DNA ladder marker and expected DNA bp for the SDC2 construct and MMP-7. We also added a sequence alignment figure between human and rat SDC2 in the supplementary.
(2) Why does the S2-FLD peptide which shows the lowest Kd (thus the greatest affinity) does not inhibit cell adhesion, but on the contrary it improves according to results in Figure 3A? This suggests that this interaction is not directly responsible for the differences in cell adhesion in Figure 3A. The authors should comment on this aspect in the manuscript. Figure 3B: It appears no difference between S2-P and any other peptide variant. (3) Also why S2-FLD was not tested? Please add S2-FLD here and to the rest of the experiments through the manuscript.
Response: Although S2-FLD has the lowest affinity, as shown in Fig. 3A, it did not reduce the attachment of HT29-SDC2 cells to the MMP-7 pro-domain. Therefore, we did not show the data in the original version of the manuscript. However, we added in the revised version of the manuscript in Figs. 4 and 5 to assess the anticancer ability against colon cancer cells, as suggested.
(4) Figure 3C: Please add some parameter (Mander's overlap coefficient or Pearson correlation coefficient) to quantify the degree of colocalization. Also a statistical test should be used to assess the differences.
Response: We deleted these data in the revised version of the manuscript and modified the text accordingly.
(5) p5, L179-182: The authors state: ‘These inhibitory activities of S2-D and S2-FE were stronger than those of S2-P …’, but the results show no changes or only modest changes compared to S2-P in Figures 3B and C. In my opinion the only real change is between no peptide and any other peptide tested (S2-P, S2-D, S2-FE). Please add real convincing results to sustain this claim or rephrase.
Response: Since the sentence is unnecessary, we deleted the sentence in the revised manuscript.
(6) p6 L202 and Figure 4: How do the authors choose the peptide concentration to test the cytotoxicity? How does this compare with the ones used for the results in Figure 3 or Figure 2? Please explain in the manuscript the rationale for this. Also, for the best performing peptide a gradient of concentrations could be tested to demonstrate the specificity of the effect.
Response: Our previous study revealed that 50 nM of S2P showed anticancer activity both in vitro and in vivo (Choi S. et al., Oncotarget 2015). We modified the text as follow: “Since the interaction of SDC2 with MMP-7 is important to regulate cancer activity [11, 26] and 50 nM of S2P showed anticancer activity both in vitro and in vivo [30], we further investigated whether 50 nM of S2-D and S2-FE could inhibit the cancer activity of HT29-SDC2 cells (Fig. 4).”
(7) Figure 4B-D and through the rest of the manuscript: Again the authors should test statistically if it is any difference between S2-P and S2-D/S2FE not between no peptide and the rest of the conditions. Otherwise they cannot claim that S2-D and S2-FE compares better to S2-P (the WT).
Response: Under our experimental conditions, we could detect enhanced anticancer activity of S2-D and S2-FE only in HCT116 cells. Therefore, in Fig. 4 using HT29-SDC2 cells, we concluded that “Unexpectedly, despite the improved interaction with the MMP-7 pro-domain, both SP-D and SP-FE showed inhibitory effects similar to that of S2-P in HT29-SDC2 cells (Fig. 4).”
(8) Figure 5C: How many cells were included for the MOC calculation? Please include a graph with values calculated for each cell analyzed and test if any significant difference can be found between S2-P and S2-D/S2-FE to verify if the two peptides indeed decrease the co-localization of SDC2 and MMP-7, compared to the initial peptide. Just declaring some values does not help to sustain the author’s conclusion.
Response: MOCs were obtained from cells in 6 randomly selected regions containing more than 30 cells and the graph was added in the revised Fig. 5E.
(9) Figure 7 and 8: Please add similar images and measurements for the animals injected with CT26-luc cells not treated with any peptide as stated in the text. Also following that mice were sacrificed the author’s should demonstrate the presence of the peptides at the end of the experiment.
Response: Since we intended to test the improved activity of the synthetic peptide compared to S2-P, we used the S2-P peptide as a control only. We also did not test for the presence of the peptides at the end of experiments. Therefore, we need to repeat the experiment to properly address the reviewer’s comment. However, since the editors only allow us 10 days for the revision, please understand the fact that I was not able to provide the requested data.
Minor points:
(1) p3L100-102: ‘In this study, we further determined whether those interaction is sufficient for their interaction…’. Please rephrase.
Response: We modified the text as follows: “In this study, we further determined whether this interaction is sufficient for cell surface localization of MMP-7 (Fig. 1)”
(2) Figure 1A: The bars represent mean raw fluorescence or normalized values? If yes to what is normalized? Please describe in the figure legend.
Response: It is described in the legend of Fig. 1A.
(3) Figure 1B: Please use a single notation through the manuscript either SDC-2 or SDC2.
Response: This has been corrected
(4) Figure 8: Missing p values and no peptide control.
Response: p-values have been added in revised Fig. 8.

Reviewer 2 Report
The manuscript by the research group of Prof. Eok-Soo Oh describes the anticancer activity of novel S2-P derivatives obtained by a few replacements of Tyr51 flanking amino acids. In particular, authors investigated the ability of these four peptides to interact with the MMP-7 pro-domain and to further inhibit the SDC2-MMP-7 interaction. In my opinion, in vitro investigations did not show an improved performance with respect to the reference compound S2-P, as differently stated by the authors. However, I do agree that in colon cancer in vivo model peptide compound S2-FE exhibited indeed more effective inhibition of primary tumor growth and metastasis than S2-P, suggesting S2-FE as one promising compound.
The number of derivatives presented in this work (4 peptides) represents a limit for the potential of S2-FE. I would have expected more changes on the S2-P sequence around Tyr51, according to the rationale of the work. For example, why just Leu and Phe as non-polar amino acids were explored on position 49? Why only Phe49-Glu52 combination instead of Phe49-Asp52, Leu49-Glu52 and many others? Authors should address this point and expand this library, prior application of further strategies that may involve replacement with unconventional amino acids and/or cyclization, aimed at improving the stability (as stated in the conclusions).
The secondary structures of peptides should be also investigated (CD or NMR techniques) to provide more information regarding the biological differences between S2-P and its most interesting derivatives.
Other issues
- I would suggest to revise the title as at the present form results to me redundant, not very informative and unattractive. Take this as suggestion.
- The English should be revised, especially in the abstract and introduction sections.
- The cited substitutions are not correct. S2-FE stems from Asp-to-Phe and Asp-to-Glu substitutions. Please, check it out.
- Page 2, line 91. “I” should be “we”.
- Quality and improvements of figures. 1) Some inconsistencies with the use of histograms: show error bars only above in histogram of 1A. Border colors of histogram 2B. Place labeling in vertical or oblique (x axis) in histograms 2B, 4A, 5A, and 5B. 2) Figure 2A: the docking structure of S2-P and PDMMP-7 can be zoomed on the site of interaction.
- Page 4, line 155-157. This sentence is ambiguous. Is the “interaction favored by the maintenance of acidic surroundings around Tyr51” despite 3 out of 4 derivatives reported Asp-to-X(non acidic) replacement? Please, authors should comment on this.
- Page 11, lines 346-360. This part does not sound in the “discussion” section.
Author Response
Reviewer 2
The manuscript by the research group of Prof. Eok-Soo Oh describes the anticancer activity of novel S2-P derivatives obtained by a few replacements of Tyr51 flanking amino acids. In particular, authors investigated the ability of these four peptides to interact with the MMP-7 pro-domain and to further inhibit the SDC2-MMP-7 interaction. In my opinion, in vitro investigations did not show an improved performance with respect to the reference compound S2-P, as differently stated by the authors. However, I do agree that in colon cancer in vivo model peptide compound S2-FE exhibited indeed more effective inhibition of primary tumor growth and metastasis than S2-P, suggesting S2-FE as one promising compound. The number of derivatives presented in this work (4 peptides) represents a limit for the potential of S2-FE. I would have expected more changes on the S2-P sequence around Tyr51, according to the rationale of the work. For example, why just Leu and Phe as non-polar amino acids were explored on position 49? Why only Phe49-Glu52 combination instead of Phe49-Asp52, Leu49-Glu52 and many others? Authors should address this point and expand this library, prior application of further strategies that may involve replacement with unconventional amino acids and/or cyclization, aimed at improving the stability (as stated in the conclusions).
Response: Since we hypothesized that the acidic residues near Tyr 51 might influence the interaction affinity based on structural information, we either added acidic amino acids or changed acidic amino acids to nonpolar amino acids near Tyr 51. Specifically, we used the Single Amino Acid Mutation Change of Binding Energy (SAAMBE) method to check the stabilizing energy score of each amino acid mutations on PPIs, and then selected amino acid sequence of the synthetic peptides. Thus, we changed the text as follows: “We generated four modified peptides from the original sequence of S2-P (Fig. 2A) based on stabilizing energy score by the Single Amino Acid Mutation Change of Binding Energy (SAAMBE) method [29].”
The secondary structures of peptides should be also investigated (CD or NMR techniques) to provide more information regarding the biological differences between S2-P and its most interesting derivatives.
Response: The secondary structure was not determined because the length of the peptide was too short.
Other issues
(1) I would suggest to revise the title as at the present form results to me redundant, not very informative and unattractive. Take this as suggestion.
Response: I prefer to leave the title as it is. Thanks!
(2) The English should be revised, especially in the abstract and introduction sections.
Response: The text in the abstract and introduction has been corrected by a professional English editor.
(3) The cited substitutions are not correct. S2-FE stems from Asp-to-Phe and Asp-to-Glu substitutions. Please, check it out.
Response: It is correct.
(4) Page 2, line 91. “I” should be “we”.
Response: This has been corrected
(5) Quality and improvements of figures. 1) Some inconsistencies with the use of histograms: show error bars only above in histogram of 1A. Border colors of histogram 2B. Place labeling in vertical or oblique (x axis) in histograms 2B, 4A, 5A, and 5B. 2) Figure 2A: the docking structure of S2-P and PDMMP-7 can be zoomed on the site of interaction.
Response: These have been corrected as suggested.
(6) Page 4, line 155-157. This sentence is ambiguous. Is the “interaction favored by the maintenance of acidic surroundings around Tyr51” despite 3 out of 4 derivatives reported Asp-to-X(non acidic) replacement? Please, authors should comment on this.
Response: We replaced Ala with acidic amino acids like Asp and Glu.

Round 2
Reviewer 1 Report
The authors have provided a revised version of their manuscript. Although they do address some of the initial concerns, I find that these were addressed only partially. I find it difficult to recommend publication considering some aspects which I find vital for the authors to sustain their conclusions. Below I have detailed some of my concerns:
Major points:
- Why do the authors delete the data in Figure 3C? In general I have asked to provide real convincing results regarding their immunofluorescence experiments, by showing the Manders overlap coefficients (MOC) distribution between the tested conditions and the statistical test used with p values calculation. Just deleting the data from the manuscript will not improve the quality of the article.
- Similar for Figure 5C, I did not find the graph which was claimed to be provided in Figure 5E, regarding MOC data.
- As mentioned earlier, a gradient of s should be tested, given the large difference between the reported Kd and the concentration used in the experiments.
- Figure 7 and 8: For the animal experiments, the no peptide ctr is missing and the authors do not provide any evidence reading the presence of the peptides. The authors could ask for an extension or could consider a possible resubmission after adding these results, to sustain their conclusions.
Minor points:
- Figure 3A and beyond: Please indicate in the figure legend the statical test and the correction used for calculating the p values.
Author Response
Reviewer 1
The authors have provided a revised version of their manuscript. Although they do address some of the initial concerns, I find that these were addressed only partially. I find it difficult to recommend publication considering some aspects which I find vital for the authors to sustain their conclusions. Below I have detailed some of my concerns:
Major points:
- Why do the authors delete the data in Figure 3C? In general I have asked to provide real convincing results regarding their immunofluorescence experiments, by showing the Manders overlap coefficients (MOC) distribution between the tested conditions and the statistical test used with p values calculation. Just deleting the data from the manuscript will not improve the quality of the article.
Response: In contrast to HCT 116 cells, which are more invasive cancer cells than HT29 (shown in Figs. 5 and 6), selected peptides (e.g. SP-D and SP-FE) showed similar inhibitory effects as S2-P in HT29 cells, which is an early stage of colon cancer cells. This was unexpected when we considered the peptide’s better interaction with the MMP-7 pro-domain (Fig. 2) and the activity to reduce the interaction of the MMP-7 pro-domain with SDC2 (Fig. 3A). Similarly, S2-D and S2-FE decreased the cell-surface level of MMP-7 (Fig. 3B) and the co-localization of SDC2 with MMP-7 (Fig. 3C, deleted in the revised manuscript) in HT29 cells comparable to that of S2-P. However, in the case of HCT116 cells, both peptides showed better inhibitory effect than S2-P (Figs. 5 and 6). To emphasize the peptide effect on HCT116 cells, and because Fig. 3B contained sufficient information for their effect on membrane localization, we deleted the old Fig. 3C in the revised version of the manuscript.
- Similar for Figure 5C, I did not find the graph which was claimed to be provided in Figure 5E, regarding MOC data.
Response: The graph in Figure 5E is from a soft agar colony formation assay. MOC data were described in the text and shown below the confocal images of Figure 5F.
- As mentioned earlier, a gradient of s should be tested, given the large difference between the reported Kd and the concentration used in the experiments.
Response: Previously, we showed that 50 nM of S2-P showed anticancer activity both in vitro and in vivo (Choi S. et al., Oncotarget 2015). To compare the inhibitory effect of the modified peptide with the S2-P under the same condition, we used 50 nM of the synthetic peptide.
- Figure 7 and 8: For the animal experiments, the no peptide ctr is missing and the authors do not provide any evidence reading the presence of the peptides. The authors could ask for an extension or could consider a possible resubmission after adding these results, to sustain their conclusions.
Response: Since the purpose of this study was to improve the activity of peptide S2-P, we designed the animal study using peptide S2-P as control without peptide control. Therefore, no peptide control is irrelevant for this study.
Minor points:
- Figure 3A and beyond: Please indicate in the figure legend the statical test and the correction used for calculating the p values.
Response: It was corrected.

Reviewer 2 Report
None
Author Response
Comments and Suggestions for Authors: None.